# Toward Versatile Small Object Detection with Temporal-YOLOv8

**DOI:** 10.3390/s24227387

**Published:** 2024-11-20

**Authors:** Martin C. van Leeuwen, Ella P. Fokkinga, Wyke Huizinga, Jan Baan, Friso G. Heslinga

**Affiliations:** TNO, Defence, Safety and Security, 2597 AK The Hague, The Netherlands; martin.vanleeuwen@tno.nl (M.C.v.L.);

**Keywords:** small object detection, YOLO, temporal object detection

## Abstract

Deep learning has become the preferred method for automated object detection, but the accurate detection of small objects remains a challenge due to the lack of distinctive appearance features. Most deep learning-based detectors do not exploit the temporal information that is available in video, even though this context is often essential when the signal-to-noise ratio is low. In addition, model development choices, such as the loss function, are typically designed around medium-sized objects. Moreover, most datasets that are acquired for the development of small object detectors are task-specific and lack diversity, and the smallest objects are often not well annotated. In this study, we address the aforementioned challenges and create a deep learning-based pipeline for versatile small object detection. With an in-house dataset consisting of civilian and military objects, we achieve a substantial improvement in YOLOv8 (baseline mAP = 0.465) by leveraging the temporal context in video and data augmentations specifically tailored to small objects (mAP = 0.839). We also show the benefit of having a carefully curated dataset in comparison with public datasets and find that a model trained on a diverse dataset outperforms environment-specific models. Our findings indicate that small objects can be detected accurately in a wide range of environments while leveraging the speed of the YOLO architecture.

## 1. Introduction

Automated object detection is increasingly relevant within various domains, including robotics [1], security [2], and defense [3]. Compared to a human operator, automated systems can improve detection speed, accuracy, and consistency. This automation hugely increases scalability and thus allows for new capabilities, including the constant monitoring of large areas, such as territorial boundaries or an extended battlefield. For defense and security applications, early detection of objects of interest is beneficial, since this translates to more time for analysis and response. Distant objects will typically only cover a small number of pixels in video and have a low signal-to-noise ratio, increasing detection difficulty [4].

In recent years, deep learning [5] has become the method of choice for various image analysis tasks, including semantic segmentation [6], classification [7], and object detection [8]. However, small object detection (SOD) remains an open challenge [4,9]. For example, most deep learning-based detectors do not exploit the temporal information that is available in video, even though this context is often essential when the signal-to-noise ratio is low. In addition, methods are typically designed around medium-sized objects, making use of intersection over union (IoU)-based losses, which are suboptimal for small objects [10]. Furthermore, datasets for the development of small object detectors are still limited in size and quality. They lack diversity and are typically task-specific, and the smallest objects are often not well annotated, if at all.

In this study, we address the aforementioned challenges, focusing on the detector model design and the use of temporal context and high-quality data to develop a high-performing small object detector that can be applied in a wide variety of environments and circumstances. We build upon the latest developments in the YOLO (You Only Look Once) [11] architecture, using YOLOv8 [12] with an extension for temporal inputs, similar to Corsel et al. [13]. We optimize a training pipeline for SOD by using bounding box augmentations and introduce a mosaicking augmentation method that carefully balances the foreground and background. Our final contribution is an investigation into the effect of having a carefully curated dataset that contains a diverse set of challenging small object examples. An evaluation is performed on a separate dataset with small objects, including those in naval, land, and military environments.

## 2. Related Work

### 2.1. Object Detection and YOLOv8

Over the past decade, numerous deep learning architectures for object detection have been introduced, such as R-CNN [14], YOLO [11], DETR [15], GLIP [16], and Grounding DINO [17]. Among these, YOLO and its variants have gained significant popularity due to their robust detection capabilities coupled with real-time performance. YOLOv8 [12] further improves on the previous versions with an anchor-free detection system, optimized convolutional layers, and advanced data augmentations, including mosaicking.

Similar to most object detectors, YOLOv8 extracts appearance (or spatial) features using only a single frame as input, discarding key temporal information. In contrast, prior to advances in deep learning, most traditional SOD methods used temporal information to detect changes and, in this way, infer object locations [18]. These methods rely on either per-pixel background intensity modeling [19] or direct frame subtraction [20,21].

### 2.2. Temporal Information and Deep Learning

The integration of temporal information into deep learning-based object detection has been shown to be promising. Heslinga et al. [4] showed that leveraging temporal information improves deep learning-based object detection, especially for small (5 to 15 pixels) and very small (<5 pixels) objects.

Recent research has increasingly focused on combining appearance and temporal features within deep learning models. For instance, long short-term memory (LSTM) networks have been used to detect moving vehicles in video streams [22] and various objects within the ImageNet VID benchmark dataset [23,24]. For the same dataset, Zhu et al. [25] obtained good detection performance by aggregating nearby features along motion paths extracted with convolutional neural networks (CNNs). Bosquet et al. [26] proposed a spatio-temporal CNN, achieving promising results for SOD.

More recently, the authors of STARNet [27] significantly enhanced SOD in video by leveraging spatio-temporal features, specifically through the use of a novel GCRU cell for feature propagation. Their approach demonstrates superior accuracy and efficiency compared to existing methods when implemented on embedded devices. He et al. [28] introduced TransVOD, incorporating temporal information into detection transformers [15] for object detection in video. Zhou et al. [29] further improved this with high performance on the ImageNet VID dataset. However, it is important to note that TransVOD was not specifically designed or tested for the detection of small objects.

### 2.3. Temporal Features in YOLO for Small Object Detection

Researchers have incorporated temporal context to further improve the performance of YOLO architectures. Luesuttiviboon et al. [30] enhanced YOLOv2 by using information from preceding frames to improve the contrast between moving targets and the background, leading to better drone detection. Alqaysi et al. [31] employed an ensemble of YOLOv4 models for bird detection in grayscale videos. The best-performing model of the ensemble used a stack of three frames to incorporate temporal information. Corsel et al. [13] presented a similar approach by developing architectural variants of YOLOv5 that allowed for the input of a stacked sequence of frames. These variants showed high potential for SOD. One of the proposed models, *Temporal-YOLO* [13], is used in this study.

### 2.4. Challenges in Small Object Detection

SOD presents unique challenges, primarily due to the difficulty in extracting accurate features from limited visual data and the scarcity of large-scale datasets specifically for small objects. Cheng et al. [32] provide an extensive overview of these challenges and of techniques developed to address them. However, their work does not focus on the use of temporal information.

One of the major challenges highlighted by Cheng et al. [32] is the smaller ratio of target (foreground) pixels to background pixels compared to standard object detection tasks, known as *foreground–background imbalance*.

Existing solutions address this challenge either through data augmentation strategies, to introduce additional small objects into the training set, or by adapting the loss function to place additional emphasis on small objects.

Dense object detectors like CenterNet [33] and FCOS [34], which predict objects at every pixel or grid point without predefined anchor boxes, have shown improved capabilities in detecting small or densely packed objects. This aligns with the challenges outlined by Cheng et al. [32], who emphasize that traditional object detection models struggle with low-quality feature representations and inadequate sampling for small objects. CenterNet, for instance, represents objects as points, predicting their centers and sizes, a concept further advanced by Zhou et al. with CenterTrack [35]. Recently, Poplavski et al. [36] won the Airborne Object Tracking (AOT) Challenge by building on these principles.

Another challenge in SOD is the sensitivity to minor deviations in the prediction of the bounding box. This leads to a significant drop in IoU compared to medium-sized or large objects [32]. This sensitivity arises because small offsets in bounding box coordinates can disproportionately affect the IoU score, particularly when the object’s surface area is minimal. These errors can stem from both model predictions and annotation inaccuracies, considering the difficulties in the precise annotation of small objects. To address this, advanced loss functions like Complete IoU (CIoU) [37] and its variants, including Soft-CIoU [38] and Wise-CIoU [39], and its combination with the Normalized Wasserstein Distance (NWD) [40] have been developed. Zhang et al. [41] used Generalized IoU (GIoU) as part of a combined CIoU loss in YOLOv5, where GIoU also focuses on non-overlapping areas in addition to the overlapping regions. While YOLOv8’s [12] IoU-based loss function has been optimized for small objects [10], issues remain for very small, moving targets. Instead of further modifying the loss function, we propose a technique to augment the bounding boxes.

### 2.5. Datasets for Small Object Detection

In object detection research, datasets play a critical role, alongside advancements in model architecture and training methodologies. The MS COCO dataset has become the standard for training most object detectors, including YOLO [12] and Centernet [33]. The ImageNet VID benchmark [23,24] dataset is commonly used for video object detectors.

In the domain of video-based SOD, however, identifying a widely accepted baseline dataset is challenging. Unfortunately, many popular datasets such as UAV123 [42], VIRAT-ground [43], Visdrone [44], and small90 and small112 [45] fail to provide sufficient examples of objects with an area of under 100 pixels. Even though these very small objects may be present in the video material, they are often not annotated. Additionally, the sequences in these datasets almost never have moving background elements, such as moving vegetation.

While typical video datasets still have shortcomings for SOD, several domain-specific video datasets exist that do sufficiently address very small objects. Some notable examples are the AOT dataset [46] from the previously mentioned Airborne Object Tracking Challenge and the Video Satellite Objects (VISO) dataset [47]. These datasets provide high-quality annotations for very small objects but exclude annotations for objects that are not relevant to their application.

## 3. Methods

As previously described, current deep learning methods still have various shortcomings. They do not exploit motion information, rely on IoU losses during training, and do not compensate for foreground–background imbalance in the training data. Several techniques are proposed to remedy these issues, which will be described in the coming sections.

### 3.1. Temporal-YOLOv8

Accurate and fast small object detection is crucial for timely decision making in defense and security applications. The YOLOv8 architecture is well suited to fulfilling these requirements, making it an excellent starting point. However, standard YOLO models are not inherently designed to utilize information from multiple frames, which limits their ability to detect small moving objects.

To exploit multi-frame context for YOLO architectures, Corsel et al. proposed the Temporal-YOLO (*T-YOLO*) concept [13], visualized in Figure 1. By default, YOLO models are trained using colored (RGB) images. However, these three channels can also be used to insert grayscale frames from three different time steps. In this way, YOLOv8 can be used with multi-frame inputs and exploit motion cues without requiring modifications to the underlying architecture. As such, Temporal-YOLOv8 has the same computational requirements as standard YOLOv8.

#### 3.1.1. Variants

Earlier work has shown that YOLO detection results improve when inputs for the model are adapted to provide more (temporal) context [13]. In this study, the number of input channels for the first convolution layer is extended to further increase the available information for the model. The first strategy, *Color-T-YOLO,* fills 9 channels with the RGB channels from 3 video frames so that both temporal and color information can be exploited by the network. With the second strategy, *Manyframe-YOLO*, the number of stacked frames is extended to 11 grayscale frames so that more advanced motion analysis can emerge from training.

### 3.2. Balanced Mosaicking

During the training of object detection models, various imbalances may occur that influence the effectiveness of the training process [48]. For SOD, foreground–background imbalance can have a particularly high impact, as small objects may only occupy a tiny fraction of an image.

An intuitive solution would be to use smaller crops around annotated objects instead of using the full-resolution frame to obtain a more balanced foreground-to-background ratio. However, this approach may reduce the variation in backgrounds present in the dataset, as a large portion is never shown to the network. While it is important that the model learns to recognize tiny objects, it should also be exposed to a wide variety of misleading and intricate background patterns. These patterns may include movement, such as wind manipulating vegetation, blowing sand, and waves in the ocean. Similarly, static backgrounds with high-frequency patterns such as forests can pose a challenge for SOD, as they may substantially diminish contrast with small objects.

To rectify the imbalance problem without sacrificing background data, we propose *balanced mosaicking*. Figure 2 provides an overview of the technique and depicts how samples for the mosaic are produced from samples out of the training dataset. Balanced mosaicking leverages crops of various sizes (420×420, 750×750, and 1920×1920) to aid the model in discerning both foreground objects and intricate background patterns. By incorporating diverse crop sizes, the model gains exposure to a wider range of spatial contexts, thereby enhancing its ability to accurately detect objects while retaining crucial information about challenging backgrounds. In Figure 3, a traditional mosaicking technique and the proposed strategy of balanced mosaicking are illustrated. The comparison reveals how the mosaicking approach can put more emphasis on the target objects rather than the background.

In addition to cropping, downscaling is another integral component of balanced mosaicking, as it helps to further reduce object sizes in the dataset. However, excessive downscaling can remove all evidence of an object’s presence. To prevent this, the scaling factor is constrained so that no object’s bounding box width or height is reduced below 10×10 pixels. (Objects in the dataset typically have bounding boxes larger than the object itself, so a minimum size of 10×10 pixels leads to sufficiently small objects.) Additionally, images that contain objects with bounding boxes already smaller than 10×10 pixels are excluded from further downscaling to ensure their features remain detectable.

Secondly, the scaling factor is constrained by the crop size required for the mosaic, ensuring that the image resolution does not drop below the target resolution. This approach naturally adjusts the downscaling factor based on the size of the crops: smaller crops (420×420 pixels) typically contain smaller, more difficult objects, while larger crops (1920×1920 pixels) include more background. If necessary, padding and cropping are applied to ensure that the cropped image fits the target resolution.

Aliasing effects can be introduced when an image is downscaled with a large factor and greatly reduced in size [49]. These effects manifest as sharp edges that would rarely be present for distant objects in real life. To enhance realism in the downscaled images and mitigate these aliasing effects, a box blur is applied before the downscaling process. The kernel size of the box blur depends on the selected scaling factor, as listed in Table 1.

### 3.3. Bounding Box Augmentations

The loss for YOLOv8 models is, to a large extent, determined by the Distance IoU (DIoU) loss, defined as
(1)$$L_{DIoU}=1-IoU+\frac{\rho^{2}\left(b,b^{gt}\right)}{c^{2}}$$
where *b* and $b^{gt}$ denote the central points of the predicted and ground-truth bounding boxes, ρ(·) is the Euclidean distance, and *c* is the diagonal length of the smallest enclosing box covering the two boxes [37]. Due to the second term in Equation (Equation 1), which is based on the distance between centers, a loss gradient can be expected, even if the predicted and ground-truth bounding boxes do not overlap. By normalizing the loss with $c^{2}$, the gradients are prevented from exploding at larger distances. However, this normalization reduces its effectiveness for SOD, as even slight deviations can saturate this loss component, resulting in very small gradients. Furthermore, as described, the YOLOv8 model’s IoU-based loss function can disproportionately penalize errors for small objects due to their small surface areas. As a result, both loss components discourage the model from detecting small objects, since slight errors in its predictions can lead to large penalties. To address this issue, bounding boxes with a width or height below 15 pixels are scaled to ensure that both dimensions are at least 15 pixels. Note that this scaling step differs from the downscaling described in Section 3.2. While downscaling reduces the size of the entire image—affecting both the object and its bounding box—this bounding box scaling is a separate augmentation. It adjusts only the bounding box dimensions without altering the image itself.

### 3.4. Metrics

Common metrics to evaluate object detectors include *recall*, *precision*, and *mean average precision* (mAP), considering an IoU overlap of at least 50% [9,50]. To make these metrics more suitable as indicators of SOD quality, we propose several modifications. Following the same reasoning outlined in Section 3.3, we consider a detection accurate if only a very small part of the bounding box overlaps with the annotation. To achieve this, we choose an IoU threshold of 1% instead of 50%. This approach still effectively measures detection performance, as the bounding boxes remain small compared to the entire frame. Furthermore, we adapt the evaluation in such a way that multiple detections within a single annotation are both accepted as correct and do not produce false positives. Conversely, one detection overlapping multiple annotations does not produce false negatives. This is illustrated in Figure 4.

With this more lenient approach, a deeper insight can be obtained into the model’s ability to direct an operator’s attention to interesting image regions rather than the model’s ability to be highly precise in its predictions. To ensure a fair comparison in the ablation study (Section 4.2), particularly for models without the guaranteed minimum bounding box size discussed in Section 3.3, we scale up detections smaller than 15×15 to at least 15×15 during evaluation.

## 4. Experiments and Results

Two sets of experiments were conducted, for which the setup and results will be described in the next sections. To demonstrate the impact of each element in our pipeline for SOD, an ablation study was performed. In addition to the efficacy of the pipeline, another study was performed in which the effect of dataset variations was measured. Through this experiment, the model’s capacity to generalize to a diverse set of circumstances was tested.

### 4.1. Datasets

Curating high-quality datasets is perhaps the most challenging aspect of deep learning-based moving object detection, which might be the reason why multi-frame small object detection approaches are not common yet. Most of today’s small object datasets are built using single images, which makes them unsuitable for detection with temporal features. While video datasets are growing in popularity, most available options at the time of writing are unsuitable for a generic deep learning small object detector. A lot of datasets only consider objects from a particular minimum size, while others are designed around a specific target class and ignore objects that are not valuable for their intended application. This leads to lower recall for tiny objects, as the amount of visual information may be insufficient to classify different targets. Public datasets also rarely feature challenging weather conditions such as wind and rain. Exposure to these situations is crucial for ensuring the model remains reliable under difficult conditions. Due to all of these factors, training models on public datasets often yields suboptimal performance in real-world applications.

To train a model that can add operational value, a dataset is required that includes a wide variety of targets and circumstances. We assembled an in-house specialized dataset named Nano-VID, which provides annotation for all moving objects captured in a wide variety of different contexts. An overview of the considered dataset characteristics is provided in Table 2. Introducing diversity in environments, targets, cameras, and viewpoints within the dataset helps prevent model overfitting and enhances its versatility. In total, the training set contains 3968 annotated frames with 12,503 annotated objects, while the test set contains 388 frames with 887 annotated objects. The distribution of object sizes in the training and test sets after processing is depicted in Figure 5. The applied processing is described in detail in Section 3.2. Notably, the 0–15 px size category has fewer samples than the other categories. This can be attributed to the lack of emphasis during annotation on perfectly fitting bounding box sizes, leading to tiny objects typically receiving a larger bounding box. As such, although the frequency of each size category is not completely equal, small objects are sufficiently well represented in both training and test sets.

#### Enhanced Annotation

Creating accurate annotations for very small objects is challenging because these objects are difficult for annotators themselves to perceive using conventional techniques. To improve visibility for small objects, an image enhancement method is employed that amplifies detected frame differences in a colored overlay for the annotator, as depicted in Figure 6. This enhancement allowed the extension of the datasets with many tiny objects that would otherwise have been left unannotated. Moreover, in instances where there is ambiguity about the existence of an object at a specific location, the uncertain object and the surrounding area are masked in black by the annotator. To further simplify the annotation process and ensure that the detector is suitable for a wide variety of use cases, we do not annotate class labels and refrain from classification during detector training.

### 4.2. Ablation Study

First, we executed an ablation study, where we trained our model using the optimal hyperparameters and then removed our techniques one by one to measure their contribution to the evaluation metrics. The ablation experiments are summarized in Table 3.

Several variations of the *YOLOv8* architecture exist that offer different trade-offs between parameter count and detection performance on the MS-COCO dataset [12]. For our experiment, pretrained weights from the *m* variant of the *YOLOv8* model were used for initialization, as preliminary experiments have shown that bigger models do not produce better detection quality on the Nano-VID dataset. Models were trained for 70 epochs using the automatic learning rate scheduler developed by Ultralytics [12] alongside the default Adam optimizer [51]. Used augmentations include Contrast Limited Adaptive Histogram Equalization (CLAHE) (p=10%), random scaling (p=10%, *scale* = 10%), and random cropping. For training and testing, the respective sets from the Nano-VID dataset described in Section 4.1 were used. Finally, to consider the variance between repeated experiments, each model was trained three times. However, the precision–recall curves for this study are based on the first run for each model. The mAP scores are reported in Table 4. The highest mAP score of 0.839 is achieved by the proposed model, while the lowest score of 0.465 is achieved by the Default-YOLO model. The precision–recall curves for each variation of the model are depicted in Figure 7 and show that the proposed model achieves the highest precision for every available recall setpoint. In fact, the same order mAP scores hold for most recall data points, as seen in Table 4. Figure 8 depicts precision–recall curves for dataloader variations. As the differences are more nuanced for this experiment, we have cropped the axes around a higher precision and recall area. The No-Mosaic model produces a curve that is well below that of the proposed model. While the Crop-Mosaic model produces similar precision values, it does have a slightly higher maximum attainable recall compared to the proposed model.

### 4.3. Dataset Impact Study

In this section, the capacity of the proposed model to generalize to a diverse set of circumstances is analyzed. Firstly, we execute a specificity study to evaluate quality differences between models trained on either a general-purpose or domain-specific dataset. With another experiment, we analyze the performance of models trained on publicly available datasets, both exclusively and in combination with Nano-VID.

#### 4.3.1. Specificity vs. Generalization

Each environment and viewpoint may present unique challenges for small object detection due to variations in object scale, context, and environmental conditions. Models that have been optimized for a particular environment are expected to outperform general-purpose models when applied to that environment. This is due to their ability to capture and optimize for the unique characteristics and constraints of the specific context, which may lead to improved accuracy. However, a diverse dataset that represents a target use case well can often be difficult to obtain, especially for small objects. In addition, developing a specialized model for each new use case may be impractical for many applications. Therefore, a generic model adapted to multiple use cases is preferable, provided that its detection accuracy can compete with that of the specifically trained model.

To evaluate the viability of this general-purpose approach, we split the dataset into subsets that represent either a Ground, Maritime, or Air-to-Ground domain. Subsequently, we conducted an experiment comparing models trained using only data from a specific domain against a generic model trained on the complete dataset. For each domain, we also extracted the corresponding test set from the Nano-VID dataset and evaluated both the specifically trained and generic models. The mAP scores for each model and domain are visualized in Figure 9. The bar plot indicates that, for each domain, the generic model either slightly outperforms or performs on par with the specifically trained model.

#### 4.3.2. Public Datasets

In Section 4.1, we mention the lack of thorough annotation of small moving objects in public datasets and propose an enhanced annotation method to improve this. However, exploiting public datasets remains an attractive proposition, as annotating all small moving objects still requires a considerable amount of effort. Although objects in these datasets are large, the preprocessing techniques introduced in balanced mosaicking (Section 3.2) could make these datasets suitable for SOD. To evaluate this approach, we assess the performance of our proposed model, trained on public datasets, when applied to Nano-VID.

For this experiment, we used two large-scale public datasets: the VIRAT-Ground Dataset [43], a dataset specifically designed for event recognition in surveillance videos, and the VisDrone-VID dataset [44], a dataset focused on drone-captured imagery. From each dataset, 2500 samples were extracted to be used during training. While the quality of these public datasets for SOD is possibly lower (e.g., increased label noise), they do offer a large amount of annotated bounding boxes to learn from, and thus, model performance could improve. To explore the effectiveness of different configurations, two experiments were performed in which models were trained based solely on a public dataset and on a combination of the public dataset and Nano-VID. For the mixed experiment, the training dataloader sampled around 80% from Nano-VID and 20% from the public dataset. To introduce more small objects, the data preprocessing steps discussed in Section 3.2 were applied. Figure 10 presents a bar plot comparing the mAP scores using the VIRAT-Ground and the Visdrone datasets. The highest mAP score is obtained when public data is omitted from the training data, while training solely on either VIRAT-Ground or Visdrone leads to the lowest mAP scores.

### 4.4. Detection Characteristics

Figure 11 contains a grid of crops containing example detections from the proposed model, intended to demonstrate its detection capacity for small objects. The targets displayed in the grid include distant people, drones, and vehicles, with some scenes captured in windy conditions. Many of the depicted targets are barely visible in a single frame, making detection challenging, even for the human eye. Yet, the model is able to detect them by exploiting movement. The detections in the grid also contain a confidence level attributed to each detection. Closer examination reveals that objects in front of or partially occluded by vegetation (such as column 0, row 4 or column 5, row 4) sometimes yield lower confidence scores compared to other detections.

## 5. Discussion

In this work, we proposed various methods to improve SOD performance with deep learning models. Firstly, we introduced the Temporal-YOLOv8 model, which captures the temporal context required to detect very small objects with few spatial cues. Secondly, we proposed a bounding box augmentation to improve the effectiveness of IoU-based loss functions for small objects. Thirdly, we incorporated balanced mosaicking to compensate for the imbalance between foreground and background samples in the dataset. Finally, we introduced enhanced annotation and scaling techniques to build Nano-VID: a video small object detection dataset with diligent annotation of all relevant tiny objects.

### 5.1. Ablation Study

By incorporating temporal context into the YOLOv8 model and applying specialized data augmentations for small objects, we increased the mAP scores on the Nano-VID test set from 0.465 to 0.839. The technique with the highest impact on the mAP score is temporal object detection, as the multi-frame model achieved a considerably higher score (mAP = 0.839) compared to the single-frame method (mAP = 0.583). The precision–recall curve in Figure 7 reveals that the multi-frame model not only detects more objects but also detects them with higher precision. As such, we conclude that multi-frame methods outperform single-frame methods, despite potential background movement, such as moving trees, plants, and waves.

In addition to the improvements from multi-frame detection, the results also reflect benefits from the various training adaptations. Notably, taking away bounding box augmentations led to a substantial 0.166-point drop in mAP, underscoring the importance of adapting the loss strategy for smaller bounding boxes. Additionally, favorable scores are also shown for balanced mosaicking, which we proposed as a method to avoid the effects of foreground–background imbalances. The benefit of balanced mosaicking becomes clear in the comparison between the proposed model and the No-Mosaic model. The No-Mosaic variant only employed full-resolution frames during training and thus contains the largest skew between the foreground and background. As evident from Figure 7, the maximum attainable recall is reduced for the No-Mosaic model, leading to a reduction in mAP from 0.839 to 0.770.

To compare the balanced mosaicking strategy with a simpler solution to foreground–background imbalance, we also tested the Crop-Mosaic model. This model uses crops centered around targets to compensate for the imbalance, but it sacrifices background examples in the process. As a result, we expected the Crop-Mosaic model to show reduced precision compared to balanced mosaicking. Supporting this hypothesis, the results show that the precision around the F1-point was considerably lower for the Crop-Mosaic model, as illustrated in Figure 7. However, the difference in mAP scores between balanced mosaicking (0.839) and fixed-size cropping (0.836) remains marginal. This can be attributed to a slightly better foreground–background ratio for the Crop-Mosaic model, giving it a slightly higher maximum attainable recall compared to the proposed model.

The Color-T-YOLO and Manyframe-YOLO models, which extend the Temporal-YOLOv8 model with additional input channels, were also evaluated in the ablation study. These techniques led to mAP scores of 0.743 for Color-T-YOLO and 0.781 for Manyframe-YOLO. This reduction in mAP can be attributed to several factors. Firstly, the absence of pretrained weights for the initial convolution layer hindered effective initialization of the training process. Secondly, the training dataset may have lacked a sufficient number of samples where these additional features could be exploited effectively. Lastly, the extra input channels might not have provided substantial relevant information from which useful features could be extracted, thereby contributing to the overall degradation in model performance.

With the ablation study, we limited the scope of our evaluation to variants of the YOLO model due to a number of factors. Firstly, there is a lack of well-established benchmark datasets available that are suitable for our use case. Additionally, in earlier work [4], we evaluated several small object detector approaches and found that Temporal-YOLO performed favorably. Finally, YOLOv8 offers strong performance, detection accuracy, and ease of use, making it a practical choice.

#### Dataset Impact Study

Two experiments aimed at establishing guiding principles for small object datasets were performed in the dataset impact study. Ideally, detectors should be able to adapt to new domains without sacrificing quality for known environments. As such, this capability was evaluated by comparing the performance of a generic model compared to environment-specific models. As evident from Figure 9, the generic model outperformed or matched the quality of the environment-specific model consistently. We conclude that a single versatile small object detector can be deployed for multiple circumstances, removing the need for specifically trained variants. Additionally, out-of-domain data may improve performance, motivating the development of an approach in which datasets are combined. However, with the public dataset experiments, we show the need for high-quality small object data, as the introduction of larger objects and/or small objects without accurate annotations clearly diminished performance, even when used in combination with Nano-VID.

### 5.2. Future Work

While the current iteration of the model performs robustly in various circumstances, there are a few areas in which improvements can still be made. We observe that when objects are fully static, the multi-frame model often performs slightly worse compared to single-frame models. It is likely that the model learns to rely on motion information for small objects during training since most objects are moving. Consequently, movement becomes a dominant feature for small objects, which appears to hinder the recall for static small objects during evaluation.

In addition to static objects, the model shows a slight bias against highly vegetated areas. While it can detect some vehicles when partially occluded by vegetation, person detection behind or near moving vegetation is not yet robust. This can be attributed to a lack of representative data, since the model frequently observes moving vegetation labeled as background during training, whereas examples of partially occluded persons are rare.

Both issues, static objects and vegetation, may be resolved by augmenting the dataset to prevent these biases and do not necessarily indicate a problem with the model’s architecture or the training approach. Nevertheless, these challenges remain interesting topics for future research.

The focus of this study is on the detection of small objects. However, operational use cases may require the detection of both small and large objects. For instance, in a military context, it can be crucial to detect both small objects, such as distant drones, and larger objects, such as vehicles or aircraft, for comprehensive situational awareness. To achieve optimal performance for such a combined detector based on temporal YOLO, balanced sampling across various size categories could be considered.

In this study, we utilize a custom-collected dataset in addition to two publicly available datasets, all of which contain only stationary-recorded videos. For many use cases, however, especially in a military context, videos will in fact be captured from moving platforms, such as flying drones or driving ground vehicles. In this case, the current setup for temporal YOLO will probably not perform as well, and motion compensation techniques should be explored.

Finally, in this study, we only focused on detection. However, classification could be relevant in many real-world applications, for instance, to distinguish drones from birds. Small objects often lack detailed features, making it difficult to classify them accurately solely based on appearance information. One way to solve this is by incorporating the tracking information, since different object classes will likely exhibit different movement patterns.

## 6. Conclusions

In this study, we introduced temporal context, augmentation methods, and sampling techniques that effectively transform YOLO into a state-of-the-art detector for very small objects. Our findings emphasize the importance of diverse and high-quality datasets to achieve success in SOD. Additionally, our results demonstrate that SOD models are generalizable across various environments and conditions and offer robustness to background motion. Figure 11 illustrates that distant and difficult-to-perceive objects can be detected using the proposed Temporal-YOLOv8 model, benefiting from bounding box augmentations, balanced mosaicking, and the Nano-VID dataset. Whereas typical object detectors only provide reliable detections when targets are already obvious to humans, the Temporal-YOLOv8 model detects objects hardly perceivable by the human eye. These features make Temporal-YOLOv8 a strong candidate for defense and security applications requiring fully automated processing, early detection, and timely intervention.

## Figures and Tables

**Figure 1 sensors-24-07387-f001:**
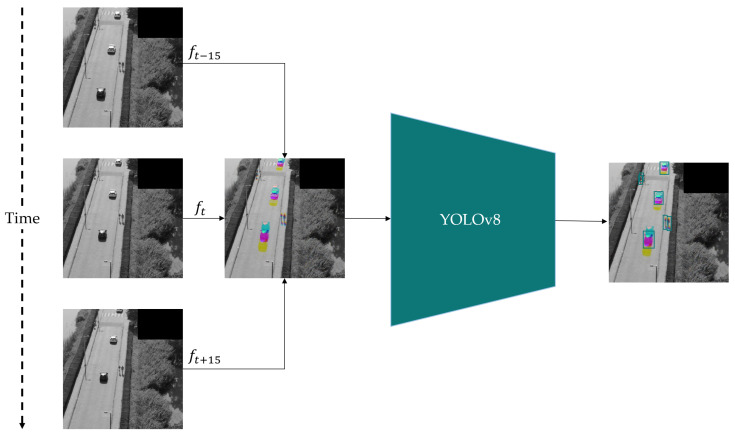
An illustration of the T-YOLO concept. Instead of using a single video frame as input, multiple frames are stacked from different time steps. In this example, the RGB channels are replaced with three gray frames. However, by slightly altering the input layer of the YOLO model, the number of frames stacked can be extended, enabling Color-T-YOLO and Manyframe-YOLO. This combined 3-channel image is provided to the model, allowing temporal context to be exploited. Assuming a 30-frames-per-second (FPS) source video, around 15 frames are sampled before and after the current frame.

**Figure 2 sensors-24-07387-f002:**
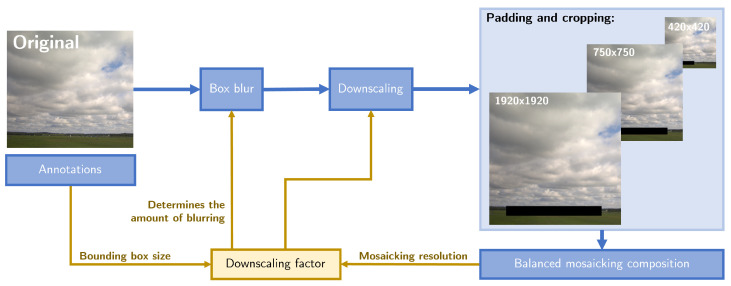
An overview of dataset preprocessing. The images are first blurred with a box blur to prevent aliasing effects. The kernel size of this blur is determined by the downscaling factor, for which the mapping is listed in Table 1. The downscaling factor itself is determined based on the size of the annotated small objects, i.e., the bounding box sizes, and the target resolution for the mosaicking composition. After blurring and downscaling, the images are padded and cropped to fully fit in the balanced mosaicking composition, for which an example is shown in Figure 3b.

**Figure 3 sensors-24-07387-f003:**
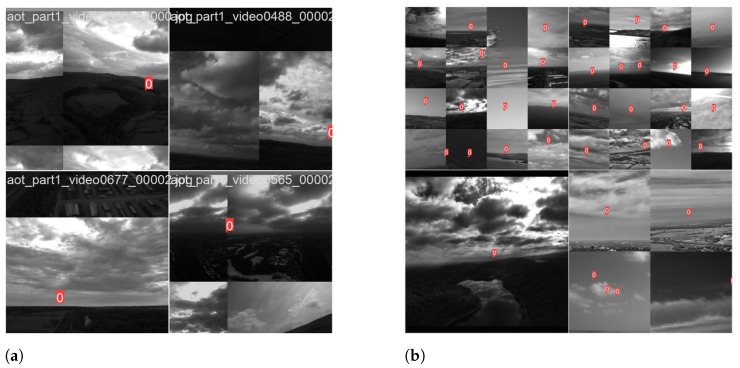
A comparison of mosaicking techniques on the Airborne Object Tracking (AOT) dataset [46]. The red annotations indicate the presence of a small object. (**a**) The built-in mosaicking in YOLOv8 with default settings. (**b**) Balanced mosaicking, where crops with varying sizes are used.

**Figure 4 sensors-24-07387-f004:**
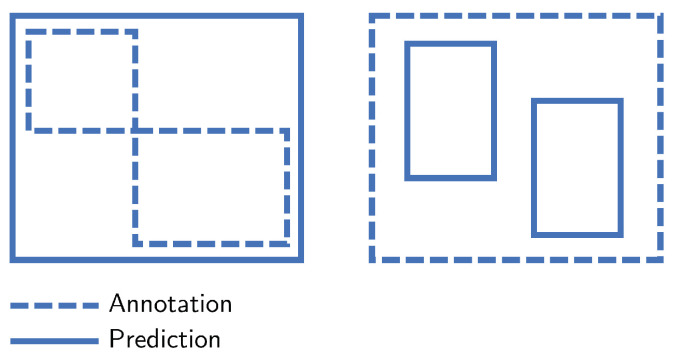
Our metric considers these examples correct, while typical object detection metrics will flag these as either a false positive or a false negative. **Left**: a single detection covering multiple annotations does not result in a false negative. **Right**: multiple detections within a single annotation do not produce a false positive.

**Figure 5 sensors-24-07387-f005:**
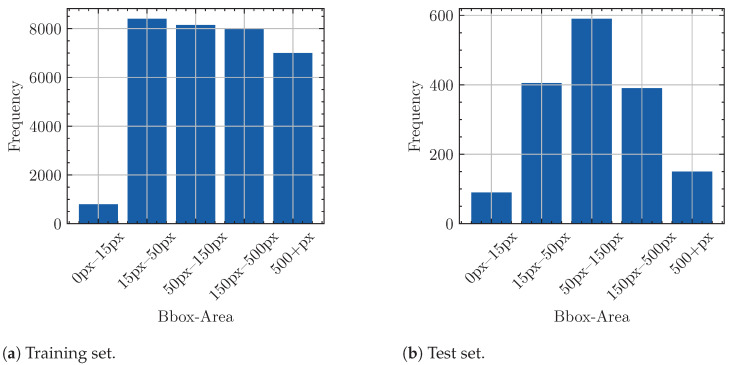
Frequency of bounding box areas found in the dataset after the processing shown in Figure 2 is applied.

**Figure 6 sensors-24-07387-f006:**
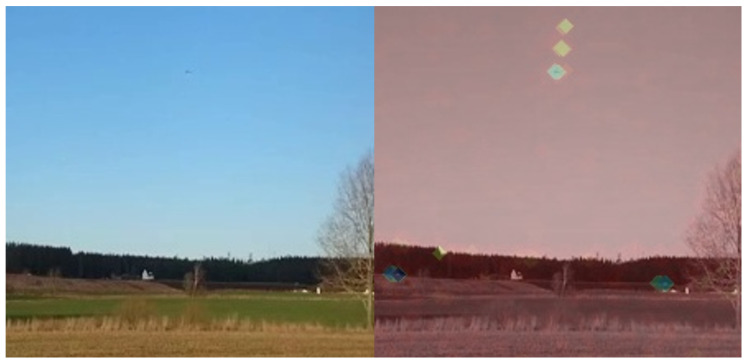
Image enhancement during annotation allows the discovery of additional tiny objects in the dataset that could otherwise easily be missed. **Left**: the original image, where the small objects are barely visible and thus very difficult to annotate. **Right**: the colored overlay based on frame differences, highlighting the small objects and facilitating more accurate annotation.

**Figure 7 sensors-24-07387-f007:**
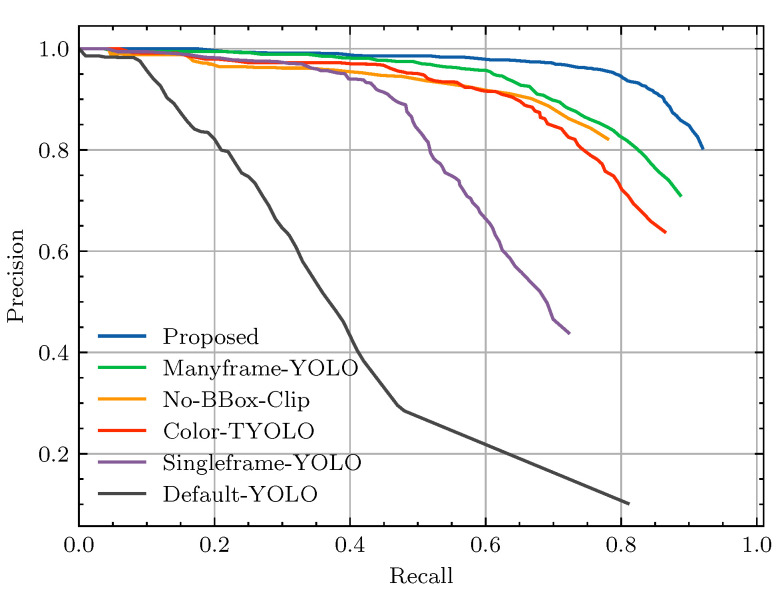
The precision–recall curves computed for each YOLOv8 variant in the ablation study based on the complete test set. For a description of each experiment, refer to Table 3.

**Figure 8 sensors-24-07387-f008:**
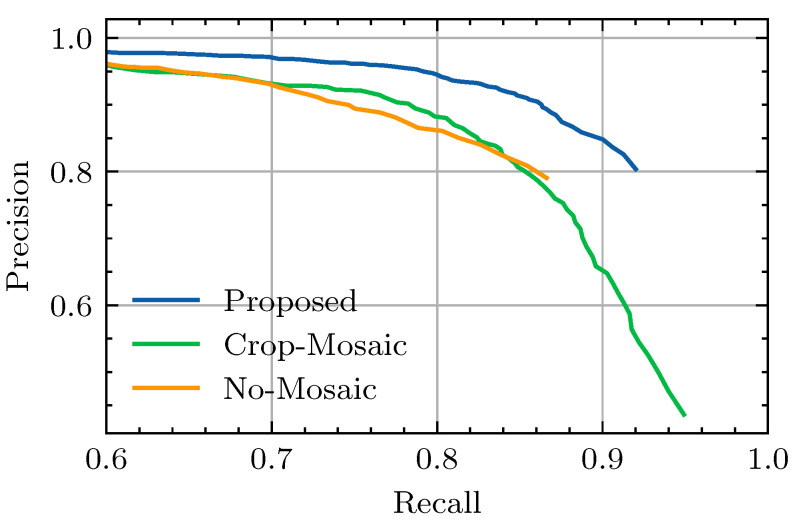
The precision–recall curves computed for each dataloader variation in the ablation study based on the complete test set. For a description of each experiment, refer to Table 3.

**Figure 9 sensors-24-07387-f009:**
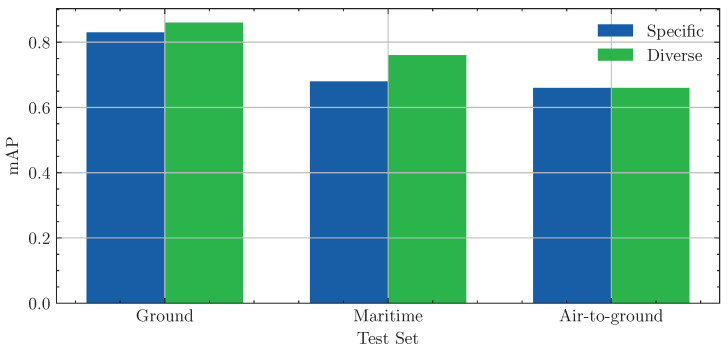
mAP scores from the specificity study. Each group of bars represents the results on a subset of the test set, while the color indicates which model was used for evaluation. The *Diverse* model was trained on the complete training dataset, while the *Specific* model was trained only on training data from the test domain.

**Figure 10 sensors-24-07387-f010:**
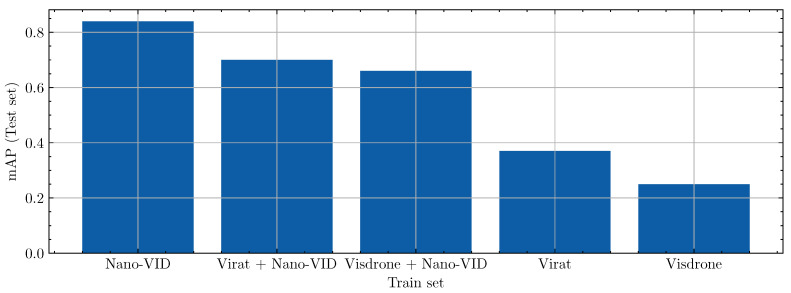
mAP scores from the public dataset study. Each bar represents the result on the Nano-VID test set based on the training set given on the x-axis.

**Figure 11 sensors-24-07387-f011:**
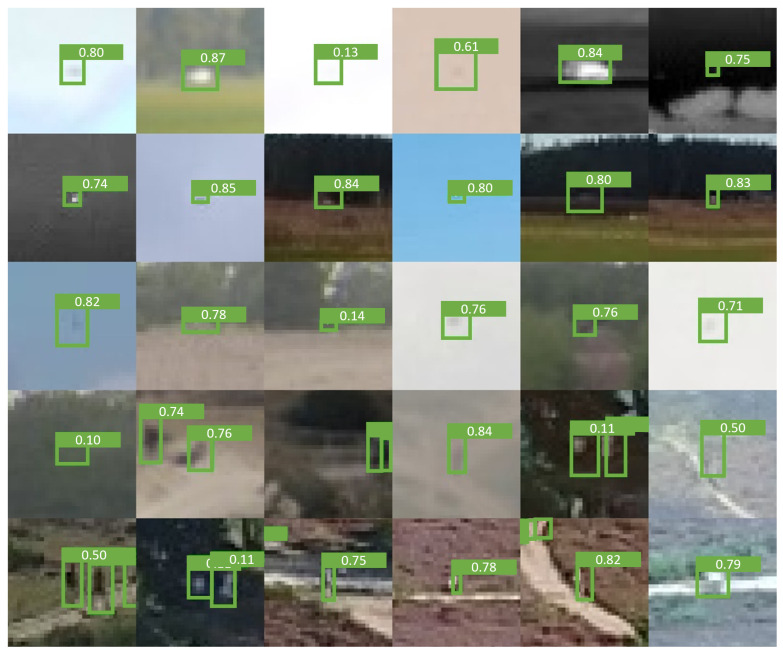
Example detections from the proposed Temporal-YOLOv8 model. These crops have been resized to four times their original size. The value drawn in each bounding box refers to the confidence of the prediction.

**Table 1 sensors-24-07387-t001:** Mapping of scaling factor to kernel size.

Scaling Factor [Range]	Kernel Size [Pixels]
0–0.15	13
0.15–0.25	11
0.25–0.50	7
0.50–0.90	3
0.9–1.0	no blur

**Table 2 sensors-24-07387-t002:** The different characteristics that were considered while assembling the datasets.

**Environments**	Forest, Desert, Plains, Urban, Port, Sea
**Targets**	Persons, Ships, Civilian and Military Vehicles, Birds, Dogs
**Cameras**	Visual and Infrared
**Conditions**	Strong Winds, Calm Winds, Rain, Sunny, Overcast
**Viewpoints**	Ground, Tower, Drone (20 m)

**Table 3 sensors-24-07387-t003:** The executed experiments in the ablation study. The precision–recall curve is computed for each variation of the model settings and used dataloaders.

Model Variations	
**Experiment**	**Description**
Proposed	Model trained using our suggested, optimally performing hyperparameters.
Manyframe-YOLO	The second model variant described in Section 3.1.1, with 11 input channels for the luminance data of 11 video frames.
No-BBox-Clip	Model trained without the bounding box augmentation, which clips the bounding boxes to at least 15 × 15, as described in Section 3.3.
Color-T-YOLO	The first model variant described in Section 3.1.1, with 9 input channels for 3 color channels of 3 video frames.
Singleframe-YOLO	Model trained using regular settings with 3 channels for one RGB frame as input but with balanced mosaicking and bounding box augmentations.
Default-YOLO	Model trained using one RGB frame as input without the data augmentation techniques presented in this work.
**Dataloader variations**	
**Experiment**	**Description**
Proposed	Model trained using the dataloader based on balanced mosaicking, as described in Section 3.2.
Crop-Mosaic	Model trained using regular mosaicking, with cropping.
No-Mosaic	Model trained using only full-resolution frames.

**Table 4 sensors-24-07387-t004:** Evaluation mean average precision (mAP) scores for the various configurations in the ablation experiment.

Experiment	mAP [std]
Proposed	0.839 [0.001]
Manyframe-YOLO	0.781 [0.001]
Color-T-YOLO	0.743 [0.001]
No-BBox-Clip	0.673 [0.003]
Singleframe-YOLO	0.583 [0.001]
Default-YOLO	0.465 [0.002]
Crop-Mosaic	0.837 [0.002]
No-Mosaic	0.770 [0.001]

## Data Availability

Data are contained within the article.

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
