# Peer review of "Toward Versatile Small Object Detection with Temporal-YOLOv8"

_sensors, 2024, doi:10.3390/s24227387_

Round 1
Reviewer 1 Report
Comments and Suggestions for Authors
In this manuscript, the authors propose adaptations based deep learning approach towards versatile small object detection with Temporal-YOLOv8.
The structure of this paper is well designed and the key idea is described by the authors. The motivation is clear, and the experimental results also verify the effectiveness of the proposed method. The performance in experiment shows the effectiveness of the proposed method. However, there are some comments shown below.
The specific comments are as follows:
Point 1: line 7, “In this study we propose adaptations to make a deep learning-based pipeline versatile for small object detection. ” => There should be a comma after study, i.e., “In this study,…”, also it is recommended to be more specific about the adaptations so that the proposed method reflects its originality and uniqueness.
Point 2: line 298, “For our experiment, pretrained weights from the YOLOv8m model are used for initialization …” , the YOLOv8m should be YOLOv8.
Point 3: line 167, 3.2. Datasets should be introduced in the Experiments Section.
Point 4: Tables 3-4 and figures 7-8, there are comparisons only the proposed model and its variation models, and there is no experimental results comparison between the proposed one and other existing approaches. Please the authors explain the reason for this.
Author Response
Dear reviewer,
Thank you for taking the time to review our work on small object detection with Temporal-YOLOv8. We appreciate you efforts in ensuring the scientific validity of our work. In the attachments you may find a pdf version of the document with the changes highlighted that we made based on your comments.
Point 1: line 7, “In this study we propose adaptations to make a deep learning-based pipeline versatile for small object detection. ” => There should be a comma after study, i.e., “In this study,…”, also it is recommended to be more specific about the adaptations so that the proposed method reflects its originality and uniqueness.
Good point, we have changed the sentence so it is grammatically correct and changed its contents so it refers to the aforementioned challenges. I believe the story in the abstract then flows well into the next ones which summarize our adaptations. You may find these changes on line 7.
Point 2: line 298, “For our experiment, pretrained weights from the YOLOv8m model are used for initialization …” , the YOLOv8m should be YOLOv8.
Thank you for pointing that out. I attempted to directly refer to the Medium or m version of the model, but I understand that this reference may be unclear. I rewrote this sentence by referring to the medium version more explicitly.
Point 3: line 167, 3.2. Datasets should be introduced in the Experiments Section.
Thanks for your suggestion, we moved this subsection to the Experiments section.
Point 4: Tables 3-4 and figures 7-8, there are comparisons only the proposed model and its variation models, and there is no experimental results comparison between the proposed one and other existing approaches. Please the authors explain the reason for this
Thank you for your valuable comment. While comparing our proposed model with other existing approaches would indeed be interesting, there is no well-established benchmark dataset available that represents our use case well. Additionally, implementing, training and evaluating alternative methods on our own dataset was unfortunately out of scope for our research. Instead, we performed an ablation study to prove the effectiveness of our methods.
That being said, we conducted an investigation of several small object detector approaches in earlier work (see reference 4), and found that temporal-YOLO performed most favorably. Additionally, YOLOv8 offers strong performance, detection accuracy, and ease of use, making it a practical choice over other detectors, which may be more computationally expensive or less compatible with tools like TensorRT.
Therefore, we focused on comparing our proposed model and its variations. We believe our experiments demonstrate the impact of our techniques and provide useful insights for future research on small object detection. We added a motivation for our choice in the discussion section on line 428.

Reviewer 2 Report
Comments and Suggestions for Authors
1. In this paper, the authors proposes adaptations to make a deep learning-based pipeline versatile for small object detection. The method can improve mAP, but what is the cost associated with it? For example, the hardware requirements, the handling frames per second?
2. How far can this proposed method apply in the real applications? What kind of research should carry on? How big is this Temporal-YOLOv8 model? Is it possible to be deployed on drones?
3.What’s your advantages compared with other temporal object detection methods? It’s hard to judge with the other methods because an in-house dataset was adopted. Is it possible to public the dataset? I believe it will be very valuable.
Author Response
Dear reviewer,
Thank you for taking the time to review our work on small object detection with Temporal-YOLOv8. We appreciate you efforts in ensuring the scientific validity of our work. In the attachments you may find a pdf version of the document with the changes highlighted that we made based on your comments.
- In this paper, the authors proposes adaptations to make a deep learning-based pipeline versatile for small object detection. The method can improve mAP, but what is the cost associated with it? For example, the hardware requirements, the handling frames per second?
Thank you for this question. The current method requires no adaptations to the YOLO architecture itself, so the same performance characteristics can be expected in terms of throughput and computation time. For a real-time system, you just have to consider that you need to buffer the frames from the camera since you need multiple frames as input for temporal YOLO. I have added a remark on line 157 to further clarify this in the paper.
- How far can this proposed method apply in the real applications? What kind of research should carry on? How big is this Temporal-YOLOv8 model? Is it possible to be deployed on drones?
Thank you for your interesting question. We measure strong performance from our methods in different circumstances, including a drone perspective. However, it should be noted that all data was captured using a fully static camera. Even the footage we used from a drone was well stabilized. To truly address the drone use-case a method must be adopted that is resilient to camera motion. This will be topic of future research. Hopefully, our extension of chapter 3.1 further clarifies what YOLO variant we use. Our goals for future research, including those concerning moving platforms such as drones are outlined in the future work section. Please let us know if you feel that we left out important research directions.
- What’s your advantages compared with other temporal object detection methods? It’s hard to judge with the other methods because an in-house dataset was adopted. Is it possible to public the dataset? I believe it will be very valuable.
Thank you for your valuable comment. While comparing our proposed model with other existing approaches would indeed be interesting, there is no well-established benchmark dataset available that represents our use case well. Additionally, implementing, training and evaluating alternative methods on our own dataset was unfortunately out of scope for our research. Instead, we performed an ablation study to prove the effectiveness of our methods.
That being said, we conducted an investigation of several small object detector approaches in earlier work (see reference 4), and found that temporal-YOLO performed most favorably. Additionally, YOLOv8 offers strong performance, detection accuracy, and ease of use, making it a practical choice over other detectors, which may be more computationally expensive or less compatible with tools like TensorRT.
Therefore, we focused on comparing our proposed model and its variations. We believe our experiments demonstrate the impact of our techniques and provide useful insights for future research on small object detection. We added a motivation for our choice in the discussion section on line 428.
I agree that it would be valuable for future research to have such a dataset open source. Unfortunately, it is not possible for us to make nano-VID publicly available due to the inclusion of confidential data. However, there are plans to release a non-confidential dataset in the near future that could be used for benchmarking temporal small object detection models.
